# A Feedback Control Sensing System of an Electrorheological Brake to Exert a Constant Pressing Force on an Object

**DOI:** 10.3390/s23156996

**Published:** 2023-08-07

**Authors:** Tomasz Spotowski, Karol Osowski, Ireneusz Musiałek, Artur Olszak, Andrzej Kęsy, Zbigniew Kęsy, SeungBok Choi

**Affiliations:** 1Celsa Huta Ostrowiec, 27-400 Ostrowiec Świętokrzyski, Poland; tspotowski@celsaho.com; 2Faculty of Mechanical Engineering, Kazimierz Pulaski University of Technology and Humanities in Radom, 26-600 Radom, Poland; k.osowski@uthrad.pl (K.O.); a.kesy@uthrad.pl (A.K.); z.kesy@uthrad.pl (Z.K.); 3Branch in Sandomierz, Jan Kochanowski University of Kielce, 25-369 Kielce, Poland; imusialek@ujk.edu.pl; 4Łukasiewicz Research Network—New Chemical Syntheses Institute, 24-110 Puławy, Poland; artur.olszak@ins.pulawy.pl; 5Department of Mechanical Engineering, The State University of New York, Korea (SUNNY Korea), Incheon 21985, Republic of Korea; 6Department of Mechanical Engineering, Industrial University of Ho Chi Minh City, Ho Chi Minh City 70000, Vietnam

**Keywords:** electrorheological fluid, electrorheological brake, strain gauge sensor, feedback control sensing system

## Abstract

The paper presents the application of a strain gauge sensor and a viscous brake filled with an electrorheological (ER) fluid, which is a smart material with controlled rheological properties, by an electric field to the fluid domain. For experimental tests, a cylindrical viscous brake was designed. The tests were carried out on a test stand especially prepared for this purpose and suitable for the examination of the impact of the rotational speed of the input shaft and the value of the electric voltage supplied to the viscous brake on pressing forces, taking into account the ER fluid temperature and brake fluid filling level. On the basis of the experimental research results, a viscous brake control system to exert constant pressing forces with feedback from a strain gauge sensor, based on the programmable logic controller, was designed and implemented. This system, using its own control algorithm, ensured a control pressing force within the assumed range, both during the constant and follow-up control. The measurement results obtained during the tests of the viscous brake designed to exert a force were presented in the form of time courses, showing the changes of the pressing force, the electric voltage applied to the brake and the rotational speed of the brake input shaft. The developed ER fluid brake control system with feedback was tested for constant and follow-up control, taking into account the impact of the working fluid temperature. During the test it was possible to obtain a maximum pressing force equal to 50 N for an electric voltage limited to 2.5 kV. The resultant error was lower than 1 N, wherein the adjustment time after changing the desired value of the force was around 1.5 s. The correct operation of both the brake and the control system, as well as the compatibility of the pressing force value and time adjustment, were determined. The main technical contribution described in this article is the design of a new type of DECPF and a new method for its control with the use of a specifically programmed programmable logic controller which simulates the proportional-integral controllers’ operation.

## 1. Introduction

In many technical devices it is necessary to exert a constant pressing force in order to, e.g., make measurements at a constant load or to obtain the required belt tension. The most used components for these purposes are pneumatic cylinders, hydraulic cylinders or mechanisms which use the pressing force of a deflected spring. However, new solutions are being sought due to the disadvantage of these methods of exerting constant pressure (e.g., the need to supply a working medium or the need to measure the spring’s deflection). The new solutions include the use of powder brakes [1], brakes with a magnetorheological fluid (MR) [2] or brakes with an electrorheological fluid (ER) [3,4].

MR and ER fluids are new materials whose rheological properties can be influenced. The fluids change their shear stress in milliseconds under an applied magnetic or electric field. This ability of MR and ER fluids allows for the control of hydraulic devices by means of changing the electric current.

The device to exert a constant pressing force (DECPF) presented in this article creates new possibilities of exerting a constant pressing force with the use of a viscous brake with an ER fluid. In this device, the pressing force is exerted on the object by a lever which is attached to a mounted brake housing. The brake’s input shaft is connected to the shaft of the electric motor. Different values of the pressing force can be obtained by changing the angular velocity of the electric motor and by applying the high voltage of the electric current applied to the electrodes of the viscous brake. Thus, the advantage of this solution is a direct use of electricity, which enables the easy implementation of digital control. A significant disadvantage of the device in some applications may be the low durability of the ER fluid. In addition, due to the fact that all the power from the electric motor is converted to heat, it is necessary to take into account the influence of temperature on the properties of the ER fluid during brake control.

Consequently, the main technical contribution described in this article is the design of a new type of DECPF and a new method for its control with the use of a specifically programmed programmable logic controller (PLC) which simulates the proportional-integral (PI) controllers’ operation.

The ER fluids used in practice are two-phase fluids (heterogenous) consisting of solid particles of the base fluid and different additives [5,6,7]. Solid particles can have different shapes, from spherical to fibrous. The outer dimensions of these particles usually do not exceed 10 μm. The particles are made of various materials, both traditional (such as starch or clay) and modern (graphene oxide or nanofibers). The base fluids are usually natural or synthetic oils [8,9,10]. Mäkelä [11] reported several dozen materials for solid particles, base fluids and additives used to produce heterogenous ER fluids. ER fluids are applied not only in viscous brakes, but also in viscous clutches [12], hydrodynamic clutches and brakes [13,14,15], combined clutches [16,17], shock absorbers [18,19], valves [20,21], cantilever beams [22], landing gear systems [23] and industrial robots [24,25].

Clutches and brakes containing a working fluid are divided into two basic groups: viscous and hydrodynamic. In viscous clutches and brakes, the driving part and driven part are connected as a result of the shear stress occurring within the working fluid. There are two basic types of viscous clutches and brakes, disc and cylindrical ones, depending on the shape of the driving and driven parts. The driving and driven parts of hydrodynamic clutches and brakes are rotors which contain blades. These parts are connected to each other as a result of the hydrodynamic impact of the working fluid on the rotor blades [26]. Tan et al. [27] presented a viscous cylinder brake with ER fluid intended to drive robots. The brake consists of the input shaft connected to the outer cylinder and the immobile output shaft connected to the inner cylinder. The outer cylinder is mounted on the output shaft with ball bearings and sealed with two sealing rings. The cylinders are electrodes connected to the poles of a high-voltage power supply. The ER fluid is selected so that in temperatures ranging from 0 °C to 80 °C the braking time would be as short as possible.

The cylindrical viscous clutch with the ER fluid, which is used to control the rotational speed of a DC electric motor, is described in the publication [28]. The brake consists of an outer cylinder and an inner cylinder. The cylinders are electrodes connected to the poles of the high-voltage power supply. The rotational speed of the brake shaft is measured with the use of a tachometer. In order to decrease the angular velocity of the electric motor, the braking torque value is changed by changing the value of the high voltage supplied to the cylinders. By using a viscous brake with the ER fluid, it is possible to obtain better results than for other control methods. Nikitczuk et al. [29] introduce a rehabilitation device which contains a cylindrical viscous brake with the ER fluid. The input shaft of the brake is connected to five cylinders placed between six fixed cylinders mounted in the casing. The movable cylinders are connected to the positive pole of the high-voltage power supply, while the fixed cylinders are connected to the negative pole of the high-voltage power supply. The tests show that using this brake results in short braking torque rise times. Tian et al. [30] describe a cylindrical viscous brake used in rehabilitation. The working gap of the brake is formed by two cylinders. During the brake tests, it is observed that the braking torque decreases due to electrical breakdowns.

Choi et al. [31] reported tests on a disc viscous brake used in a small mobile robot. Three discs are mounted on the brake’s input shaft. Between these discs and four discs there are six gaps. The design of the viscous brake obtained as a result of immobilizing the output shaft of a cylindrical viscous clutch with five gaps is shown in [26]. A disc viscous brake obtained as a result of immobilizing the output shaft of a viscous clutch is presented in [32]. Ten discs are mounted on the input shaft of the brake. The following nine discs, each of the same thickness, are connected to a fixed output shaft. The shafts are mounted in the casing. Between the discs there are gaps filled with an ER fluid.

Olszak et al. [33] and Żurowski et al. [34] described a viscous disc brake with the ER fluid being used in the gripper of an industrial robot. The brake consists of six discs mounted on the input shaft and five discs mounted on the immobile casing. The positive pole of the high-voltage power supply is connected to the clutch shaft, while the negative pole is connected to the brake shaft. Kęsy et al. [14] present a hydrodynamic brake with an ER fluid. Clutches and brakes with ER fluids are controlled by feedback control systems. Nakamura et al. [35] propose a semi-active variable viscous control method to control the transmitted torque of a cylindrical clutch with the ER fluid; the clutch is used in a robot. The value of the torque transmitted by the clutch is influenced by two quantities: the angular velocity of the input shaft of the clutch and the dynamic viscosity of the ER fluid. The ER fluid used in the clutch is a homogeneous fluid whose rheological properties (as opposed to heterogenous fluids) can be described by the Newtonian fluid model. Therefore, it is assumed that the influence of the electric field on a homogeneous fluid changes its dynamic viscosity. To control the viscosity changes, an impedance control method is used. The designed closed-loop control system is tested on a test bench where the clutch serves as a brake. The torque is measured with the use of a load cell sensor, pressed by a lever connected to the output shaft of the brake. The resultant error (which is the difference between the desired torque and the measured torque) is connected to the input of the PI controller. The output signal of the high-voltage power supply (in the form of high-voltage electricity) is supplied to the brake electrodes.

In a rehabilitation device with a cylindrical viscous brake with ER fluid [36,37], two different closed-loop hybrid controllers are used. They consist of a typical controller, an inverse model and a sliding mode with a gain scheduling component. The goal of the first control system is to maintain a constant torque by changing the voltage applied to the brake electrodes. The control system of the constant torque has two types of feedback. The primary feedback comes from the torque, the secondary feedback comes from the angular velocity, whose value is introduced into the inverse model. The device uses a PI controller; the resultant error is applied to the regulator’s input. In order to improve the parameters of the control system, a dynamic modification of the PI controller is applied. The signal coming from the PI controller is summed up with the desired torque and the signal from the adaptive controller. The value of the PI control signal is summed up with the output signal from the adaptive controller and then applied to the inverse model along with the angular velocity of the viscous brake shaft (measured by the encoder). The inverse model converts the signal of the desired torque into the value of the electric voltage supplied to the brake electrodes. Regardless of changes in the torque transmitted by the brake, the task of the second control system is to maintain a constant speed of the viscous brake shaft by changing the electrical voltage applied to the brake electrodes. The resultant error is corrected before being applied to the proportional-integral-derivative (PID) controller. The correction is made to reduce the overshoot value, and it consists in multiplying the resultant error by a constant from the defined assignment table. The output signal from the PID controller is summed with the output signal of the adaptive controller. On the basis of the inverse model, the resulting signal is converted into the value of the electric voltage supplied to the electrodes of the brake with ER fluid. The test results show that both control systems provide the required control accuracy.

Choi et al. [28] reported on the method of regulating the rotational speed of a DC motor with the use of a cylindrical viscous brake with a heterogenous ER fluid. The DC motor is connected to the shaft of a viscous brake with ER fluid via the shaft on which a tachometer and a torque sensor are mounted. The DC motor is powered by a constant, time-invariant electrical voltage. The rotational speed of the electric motor is controlled by changing the high voltage supplied to the cylinders of the brake with ER fluid with the use of a specifically programmed PC which simulates the regulator’s operation. The control system used here is a closed-loop control system with feedback from angular velocity. Two types of controllers are tested: a PID and a sliding mode controller. A smaller control error is obtained when the sliding mode controller is used. Two controlled viscous clutches with ER fluid (serving also as brakes) are used to control the angular velocity of the rotor and the drum of a rotary washing machine [38]. It is assumed that the speed of the rotor and the drum will be controlled with feedback and will be carried out independently. The PC which is used for control is equipped with measurement cards and C/A output cards used for communication with the high-voltage power supply. A PID controller is used with the settings adjusted according to the Ziegler–Nichols method. The angular velocities are measured with the use of digital inductive sensors. The resultant error supplied to the PID controller is the difference between the desired angular velocity and the measured angular velocity.

## 2. Device to Exert a Constant Pressing Force

The construction scheme of the mechanical part of the DECPF is shown in Figure 1, and the construction scheme of the cylindrical viscous brake is shown in Figure 2. A DECPF consists of a cylindrical viscous brake with ER fluid and a lever mounted on its output shaft; the lever exerts pressure on the object by means of the sensor placed at the lever’s end. Two coaxial metal cylinders form the driving part, and three coaxial metal cylinders form the driving part of the viscous brake. The driving part of the viscous brake is connected to the input shaft driven by the electric motor, while the driven part is connected to the output shaft of the brake on which the lever is mounted. These shafts are mounted in external supports to prevent solid particles of ER fluids from entering the bearings. The dimensions of the DECPF are shown in Table 1. The DECPF uses ERF#6 heterogeneous fluid made in the Department of Inorganic Chemistry and Solid State Technology Faculty of Chemistry at Warsaw University of Technology. The data of the ERF#6 fluid are shown in Table 2.

The viscous brake operates in such a way that the torque is transferred from the driving part to the driven part by friction, which results from the shear stresses occurring in the ER fluid. The value of the shear stresses can be changed by changing the electric field strength. The electric field is generated between the cylinders (which are also electrodes) of the driving and driven part. The cylinders are fixed on the shafts by sleeves made of plastic and, thus, are isolated from each other.

An increase in the high voltage applied to the electrodes causes an increase in the electric field strength. This, in turn, causes an increase in the shear stress in the ER fluid and, thus, an increase in the transmitted torque. The pressure of the end of the lever on the object increases as a result of the increase in the torque value. The pressing force is measured by a force sensor. The temperature of the ER fluid is measured using a temperature sensor located in the brake filler cap (Figure 2). To maintain the pressing force at the predetermined level, a feedback control system with a PLC is used. The high-voltage electrical supply to the cylinders is carried out by carbon-copper brushes and copper slip rings (Figure 3). This solution is possible due to the fact that the viscous brake casing, in which the filler plug is located, does not rotate.

## 3. Test Bench

Experimental research on the DECPF is carried out on a test stand specially built for this purpose. It consists of an electric drive motor, a high-voltage power supply, a PC and a control system using a PLC (Figure 4). The test bench enables the measurement and recording of the following physical quantities in time *t*: the pressing force *F*, the angular velocity of the electric motor *ω*, the control voltage *U* (supplied from the high-voltage power supply), the leakage current *i_g_* and the ER fluid temperature *T*. The measurement results are recorded with the use of a computer measurement system containing a PC with real-time measurement data recording software. The basic data of the electrical and electronic components used to construct the test bench is presented in Table 3. Installing the SM-Profinet card in the inverter renders it possible to exchange data with a factory-equipped PLC [39,40]. PLC communication with the high-voltage power supply is carried out using the Profibus protocol, due to the availability of this option in the FUG brand high-voltage power supply. The force and temperature were measured with the use of measurement cards, which are an extension of the PLC controller. The integration of various electronic components forced a modular construction of the control system.

## 4. DECPF Characteristics Research

The first stage of the bench tests is to find the relationship between the exerted pressing force *F* and the angular velocity *ω* of the input shaft of the viscous brake with ER fluid, for high voltage *U* = 0 and *U* ≠ 0. During the study of these relationships, the angular velocity *ω* and the high voltage *U* are changed in time linearly and stepwise. Then, tests are carried out to find how the temperature of the ER fluid and electrical breakdowns affect the characteristics of the DECPF.

### 4.1. Testing of DECPF Characteristics in the Absence of High Electrical Voltage

In order to obtain a linear change in the angular velocity *ω*, the velocity is increased every 5 rad/s and kept constant. This allows the inertia effect to be reduced. Due to a significant increase in the temperature of the working fluid above an angular velocity of 80 rad/s, the tests are carried out in a range from 0 to 80 rad/s. The electric motor is controlled by the inverter so as to obtain the required constant angular velocity *ω* in a short time.

The performance of the DECPF is significantly affected by the dynamic viscosity of the ER fluid. Choosing a low viscosity of ER fluid allows for a wide control range for the DECPF, because the smallest value of the pressing force *F*, which occurs for *U* = 0, depends on the viscosity. Figure 5 presents for the DECPF with ERF#6 fluid the course of changes in time *t* of: the angular velocity *ω*, the pressing force *F* and the temperature *T* of the ER fluid. Figure 5 shows that the pressing force *F* increases with increasing angular velocity ω, while the ratio of angular velocity to force *ω*/*F* ranges from 15 (rad/s)/N to 20 (rad/s)N. The reason for the deviations in the linear dependence of the force *F* on the angular velocity *ω* (especially prominent for higher angular velocities *v*) is the increasing temperature of the ER fluid.

Figure 6 shows the waveforms of the pressing force *F* for step changes in the angular velocity *ω* of the input shaft of the viscous brake with ER fluid. Due to the fact that it is not possible to change the angular velocity ω in an infinitely short time, Figure 6 shows the actual course of the angular velocity of the input shaft, measured with the use of a rotary encoder. Fluctuations in the pressing force *F* at 60 rad/s may be caused by phenomena occurring for higher angular velocities during the acceleration of the brake discs in a very short time (0.2 s). Slippage may occur in the boundary layer between the discs and the ER fluid. Then, the angular velocity of the ER fluid close to the disc surface is lower than the disc’s angular velocity. In addition, during the acceleration of the discs to a higher angular velocity, the centrifugal force acting on the solid particles is greater, which may cause local changes in the composition of the ER fluid.

The amplification factors (steady-state gain factor *k_ω_* and time constant *T_ω_*) are determined by treating the waveforms of the force *F* for individual angular velocities (as shown in Figure 6) as the response of the first-order inertial system. The time constant *T_ω_* is determined on the basis of the time for which the value of the pressing force is 0.63 of the maximum pressing force *F* value [35,41]. The values of the identified time constants *T_ω_* and steady-state gain factors *k_ω_* arepresented in Table 4.

### 4.2. Testing of DECPF Characteristics in the Presence of High Voltage

During the study of the DECPF characteristics for the linearly increasing voltage *U*, the angular velocity *ω* of the input shaft of the viscous brake with ER fluid is set at 10 rad/s and subsequently increased by another 10 rad/s up to 100 rad/s. For each constant angular velocity, the electric voltage *U* is increased and kept at a constant level from 0 kV to 3.5 kV every 0.1 kV. After reaching the maximum value of the voltage *U* equal to 3.5 kV, the value of the electric voltage is lowered every 0.1 kV down to zero. The adopted method of conducting the research allows us to limit the inertia effect. Figure 7 shows changes in the angular velocity *ω*, high voltage *U* and pressing force *F* in time *t* for a temperature *T* of 50 °C. As shown in Figure 7, for each constant value of the angular velocity *ω*, the pressing force F increases in proportion to the value of the applied voltage *U*. Moreover, for each constant angular velocity *ω*, the local minima of the force *F* occur for high voltage *U* equal to zero. The local maxima of the pressing force *F* occur for the maximal values of high voltage *U*. The increase in pressing force Δ*F* for different angular velocities *ω* (as determined on the basis of Figure 7) caused by the occurrence of high voltage *U* = 3.5 kV, is shown in Table 5. On the basis of Table 5, it is noticeable that the increase in the pressing force Δ*F* is greater the smaller the angular velocity *ω*.

Figure 8 shows the value of the pressing force *F* for step changes in high voltage *U*. Similarly to the angular velocity *ω*, it was not possible to obtain a step change in the electric voltage *U* in a very short time, so Figure 8 shows the actual course of changes in the electric high voltage *U*. Assuming (similarly to the angular velocity jump) that the waveforms of the force *F* shown in Figure 8 are the responses of the first-order inertial system, the time constants *T_U_* and steady-state gain factors *k_U_* are determined. The time constants *T_U_* are determined on the basis of the time for which the value of the pressing force is 0.63 of the maximum pressing force *F* value. The values of the identified time constants *T_U_* and steady-state gain factors *k_U_* are presented in Table 6. During the identification it is also estimated that the time constant *T_a_* of the electric power supply is 0.18 s for a voltage spike of 2.5 kV, while the steady-stage gain factor is 0.7 kV/V.

### 4.3. Research on the Influence of ER Fluid Temperature on DECPF Characteristics

The influence of the working fluid temperature *T* on the pressing force *F* is determined for two temperature ranges: from 25 °C to 38 °C and from 35 °C to 45 °C. Due to the intense heat release during the operation of the viscous brake with ER fluid, it is not possible to maintain a constant temperature for a long time during the measurements. These ranges and the maximum angular velocity at which the tests are conducted (60 rad/s) are determined on the basis of preliminary research. When the ER fluid reaches the lower temperature of the range (and after selecting a constant rotational speed), the applied high voltage *U* is increased every 0.1 kV from 0 kV to 3.5 kV. The high voltage *U* is maintained at a constant level for several seconds in order to stabilize the measurement of the force *F*. The measurement series is finished when the working fluid temperature reaches the upper value of the range. The dependence of the pressing force *F* on the electric voltage *U* for selected temperature ranges is shown in Figure 9. For all the considered angular velocities, an increase in the ER fluid temperature from the range 25 °C to 38 °C to the range 35 °C to 45 °C causes an increase in the pressing force *F*. This is connected to the increase in shear stresses.

Figure 9 shows fluctuations in the pressing force *F* for voltages *U* over 3 kV, which are caused by electrical breakdowns.

### 4.4. Research on the Impact of Electrical Breakdowns on the DECPF Characteristics

Electrical breakdowns can cause the destruction of solid particles in the ER fluid as a result of a step increase in temperature in the area of the electric spark flashover. Factors which increase the risk of electrical breakdowns might include high temperature of the ER fluid, high surface roughness of the electrodes and humidity and contamination of the ER fluid [42]. Occurrences of electrical breakdowns in the tested viscous brake cause a step increase in the electric leakage current *i_g_* flowing through the ER fluid within the working gaps of the brake. When the current exceeds the 100 mA limit of the power supply value, there is a decrease in the value of the electric high voltage and, subsequently, a decrease in the electric current intensity. After that, the voltage is automatically restored by the control system of the high-voltage power supply to the desired value. The cycle is repeated, which results in a series of breakdowns. During the initial DECPF research it was noticed that the reason for the increase in frequency of electrical breakdown occurrences may be that the ER fluid does not fill the entire volume of the viscous brake. After emptying the brake, electrical breakdowns occur at a voltage below 3.0 kV. When the brake is filled with the working fluid, the breakdown voltage exceeds 3.5 kV. In order to confirm the occurrence of this phenomenon, the brake is filled to 90 % when operating with the use of working fluid. For a constant angular velocity *ω* increased every 5 rad/s, the high voltage *U* is also increased every 0.1 kV up to 3.5 kV. The constant value of the voltage *U* is maintained for several seconds in order to stabilize the voltage and the measured pressing force.

The temperature *T* of the ER fluid is also measured. The course of the voltage *U* in time *t* is shown in Figure 10. Table 7 shows the values of the electric voltages *U_b_* for which the electric breakdowns occurred, as well as their respective pressing forces *F* for constant angular velocities. The values that have a random nature are read from charts made for 90% brake filling. Table 7 shows that both the breakdown voltage *U_b_* and the pressing force *F* increase with the increase in rotational speed ω of the input shaft of the viscous brake with ER fluid. The increase in the value of the electric voltage *U_b_* (at which breakdowns occur) with the increase in the angular velocity ω can be justified by the centrifugal force. The centrifugal force pushes the air bubbles towards the rotation axis via the ER fluid, whose viscosity is much higher than the viscosity of air. In this way, within the working gaps there is less air which has a resistance lower than the resistance of the ER fluid. That renders it difficult for the spark-over to pass through the gaps. The analysis of the temperature course *T* does not indicate that the temperature of the ER fluid increases significantly due to electrical breakdowns, which is a result of the high thermal inertia of the viscous brake with ER fluid.

## 5. Pressing Force Control System

Based on the analysis of the literature and the obtained DECPF step responses, it is assumed that the control of the pressing force is implemented in a closed-loop control system by means of changes in the electric high voltage *U* for selected constant values of the angular velocity *ω_F_*. After the preliminary tests, it is assumed that the values of the angular velocity *ω_F_* depend on the desired pressing force, as shown in Figure 11. Figure 11 shows that the ranges of the pressing force *F_d_* partially overlap. This means that the change in the rotational speed *ω_F_* from one of the selected values to another is less frequent. In order to limit the inertia effect, it is assumed, based on preliminary tests, that the change in angular velocity from one selected value to another occurs with an acceleration of 20 rad/s^2^. The scheme of the DECPF control system is shown in Figure 12.

The closed loop pressing force control system uses a digital regulator which contains proportional and integral control terms. The regulator does not contain the differentiator term because, according to publications [35,36], its operation is not favorable in systems with high inertia. The PI controller for the developed control system (Figure 12) is obtained by programming the PLC. The force *F* is measured with the use of a load cell sensor, pressed by a lever connected to the output shaft of the brake. The resultant error *e* is the difference between the desired force *F_d_* and the measured force *F*. The resultant error *e* is connected to the input of the PI controller. The output from the PI controller is the control quantity *u* described by the relation:(1)$$u=k_{P}e\left(t\right)+k_{I}\int_{0}^{t_{0}}e\left(t\right)dt$$
where *k_P_* is the steady-state gain proportional coefficient and *k_I_* is the integral factor.

The control quantity *u* is applied to the high-voltage power supply’s input. The output signal *U* of the high-voltage power supply is supplied to the brake, along with the angular velocity *ω* of the viscous brake shaft.

While choosing the coefficients *k_P_*, *k_I_* of the PI regulator, two heuristic methods developed by Ziegler and Nichols are taken into account. However, the PI controller settings obtained by these methods needed to be corrected using test-based manual selection [43,44]. The values of the steady-state gain coefficients *k_P_* and *k_I_* are presented in Table 8.

In order to assess the operation of this control system, the value of the desired force *F_d_* is increased from 10 N to 60 N (in steps of 5 or 10 N), then kept constant, and then decreased. The time of maintaining a constant value of the set force *F_d_* is selected in such a way as to achieve the control goal for an acceptable duration of the experiment. The change in the *F_d_* desired value to the next fixed value depends on the regulation time of the measured force *F*. Figure 13 presents an example of the DECPF control system’s operation for *k_P_* = 0.05, *k_I_* = 0.2 and *T* = 50 °C. Figure 13 shows that the desired force *F* is not reached for the angular velocity *ω* of 60 rad/s. Increasing this force as a result of increasing the high electric voltage *U* is not possible due to the electric breakdowns occurring more often as the temperature increases (especially when the temperature of the ER working fluid exceeds 40 °C). In order to obtain the desired course of the force *F*, it is assumed, on the basis of the tests, that if the high voltage *U* ≥ 2.5 kV then the angular velocity *ω_F_* is increased with an acceleration of 2.5 rad/s^2^ from zero up to 10 rad/s. The scheme of the thus corrected control system is shown in Figure 14. Further research shows that the modified control system (whose scheme is shown in Figure 14) allows us to obtain the desired value of the force *F_d_* for ER fluid temperatures over 40 °C, because then the high voltage reaches 2.5 kV for a time not longer than 1 s.

## 6. Discussion of Research Results

The lack of repeatability of the exerted pressing force for identical values of the angular velocity *ω* and for identical values of the high voltage *U* is caused by the random nature of electrorheological phenomena occurring within the ER fluid, and by the temperature changes. These are the reasons why a complex feedback control system is required to maintain a constant pressing force *F*. Table 4 and Table 6 show that the time constants *T_ω_* and *T_U_* have similar values. However, this is due to the fact that the brake power in the tested range does not exceed 1.2 kW, while the power of the test bench drive motor is 5.5 kW. It should be assumed that in the DECPF used in practice, the differences between the brake power and the electric engine power will be significantly smaller. This means that the pressing force *F* changes would be achieved faster by changing the value of the electric high voltage *U* than by changing the angular velocity *ω*. For this reason, the selected control variable is high voltage *U*.

In the developed control system, the maximum values of the resultant error *e* (reaching 50 rpm) occur only when the angular velocity *ω_F_* is changed. During the adjustments, the time in which the rotational speed reaches the desired angular velocity *ω_F_* ranges from 0.8 s to 0.95 s. The stepwise desired angular velocity *ω* is stabilized by the inverter after ca. 1 s, due to the necessity of limiting the torque shock changes in the electric motor. The average absolute value of the difference between the desired and measured voltage during the adjustment is 17 V, while during the steady-state operation of the power supply it is lower than 0.5 V. During the pressing force control, the resultant error *e* is lower than 1 N, wherein the adjustment time after changing the desired value of the force *F_d_* is around 1.5 s. Considering that in the control system (Figure 12) there is a dependence of the angular velocity *ω_F_* on the desired pressing force, it is possible to obtain the maximum pressing force (50 N) for the electric voltage limited to 2.5 kV. The applied control adjustment (Figure 13) consisting in increasing the speed *ω_F_* at the maximum electric voltage (2.5 kV) allowed us to reach the desired pressing force *F_d_*_,_ despite the occurrences of electrical breakdowns within the range of higher temperatures. The obtained courses of the measured physical quantities indicate that the components of the electric motor control system and the high-voltage power supply for the DECPF control system were selected correctly.

The pressing force value *F*, in the absence of electric voltage *U*, is proportional to the angular velocity *ω*. Its maximal value measured during the tests is 44 N for a rotational speed of 130 rad/s. Applying electric high voltage *U* to the electrodes of the viscous brake with ER fluid causes an increase in the pressing force *F*. The lower the angular velocity *ω*, the greater the increase in the pressing force *F*. For a rotational speed of 10 rad/s, the increase in the pressing force *F* caused by applying an electric voltage *U* = 3.5 kV is 500%, while for an angular velocity of 60 rad/s it is 35%. The temperature of the ER fluid significantly influences the value of the pressing force *F*. It was found that maintaining the required temperature of the ER fluid in the range of 45 °C to 55 °C is possible for an angular velocity of *ω* < 60 rad/s.

## 7. Conclusions

On the basis of the results of the DECPF tests, the following conclusions were reached:The random nature of changes in shear stresses within the ER fluid, the significant influence of temperature on the value of these stresses and the occurrence of electrical breakdowns are the reasons why it is necessary to use a feedback control system. The developed feedback control system with the PI controller ensured a controlled pressing force in a wide range could be obtained.Due to the intense heat release during the operation of the viscous brake with the ER fluid and due to the increase in the brake temperature (which causes changes in the shear stresses of the ER fluid), it would be beneficial to use an additional brake cooling system in order to stabilize the temperature.When designing the DECPF, it should be taken into consideration how the filling degree of the viscous brake with the ER fluid influences the occurrence of electric breakdowns. All working gaps should be completely filled with ER fluid, which would increase the stability of the control system operation.Further work on the development of the DECPF structure should focus on designing new control methods to shorten the control time in which several applications of ER fluids associated with proper control methods could be realized in practice.

It is finally worth noting that the number of recent research works on application systems or devices utilizing ER fluid is considerably less than those on application systems or devices using MR fluid. The causes may stem from several issues, including the lack of commercial products using ER fluid, the lower field-dependent yield shear stress of ER fluid itself compared to MR fluid, the high input voltage required to activate chain-like structures of the particles and so forth. This research trend can provide new inspiration or creative insight to many scholars or potential readers for successful practical realization in the near future.

## Figures and Tables

**Figure 1 sensors-23-06996-f001:**
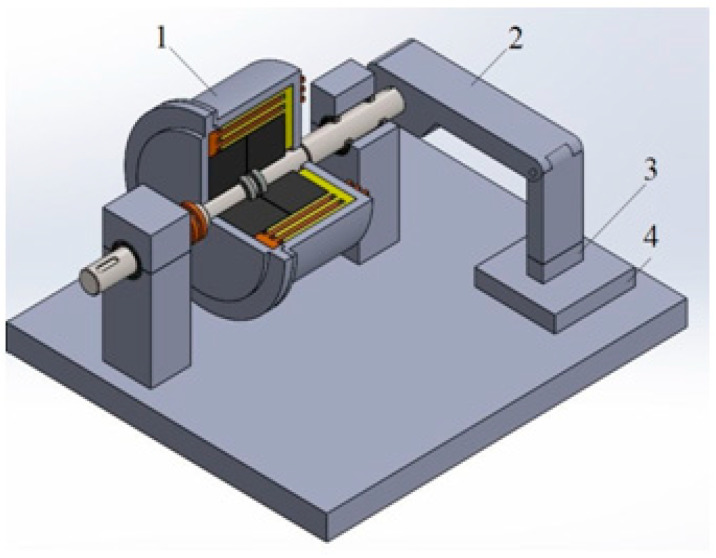
DECPF construction scheme: 1—cylindrical viscous brake, 2—lever, 3—force sensor, 4—object.

**Figure 2 sensors-23-06996-f002:**
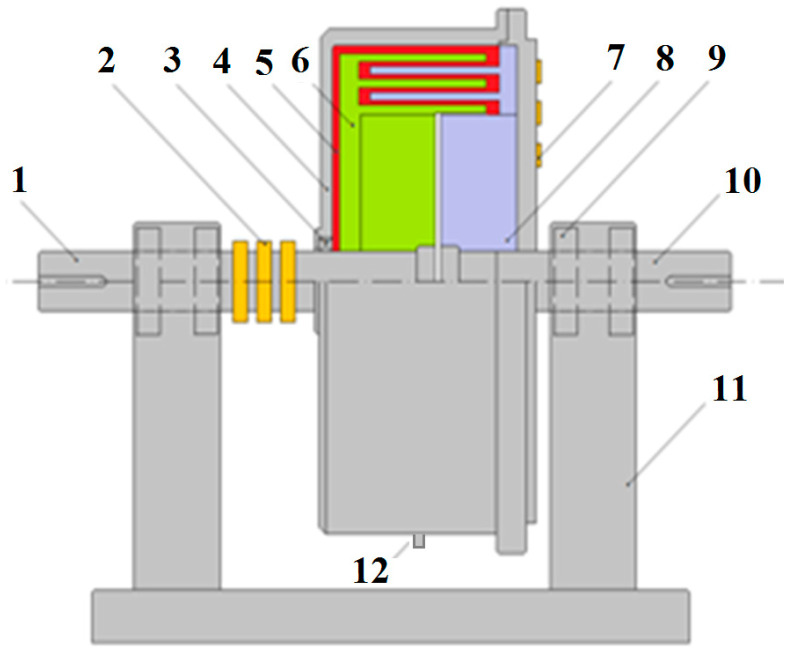
Construction scheme of a cylindrical viscous brake: 1—input shaft, 2—slip rings, 3—sealing ring, 4—housing, 5—ER fluid, 6—cylinders connected to input shaft, 7—slip rings, 8—cylinders connected to output shaft, 9—bearings, 10—output shaft, 11—supports, 12—temperature sensor.

**Figure 3 sensors-23-06996-f003:**
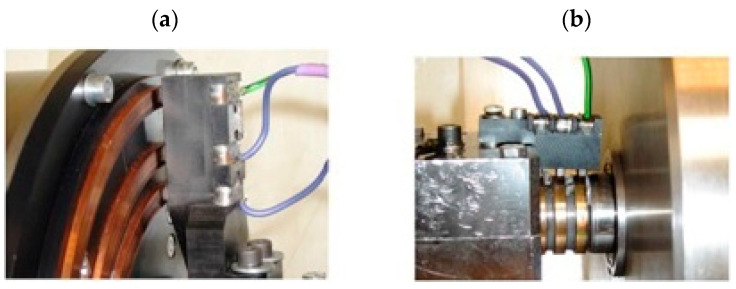
Slip rings and brushes: (**a**)—the driving part, (**b**)—the driven part.

**Figure 4 sensors-23-06996-f004:**
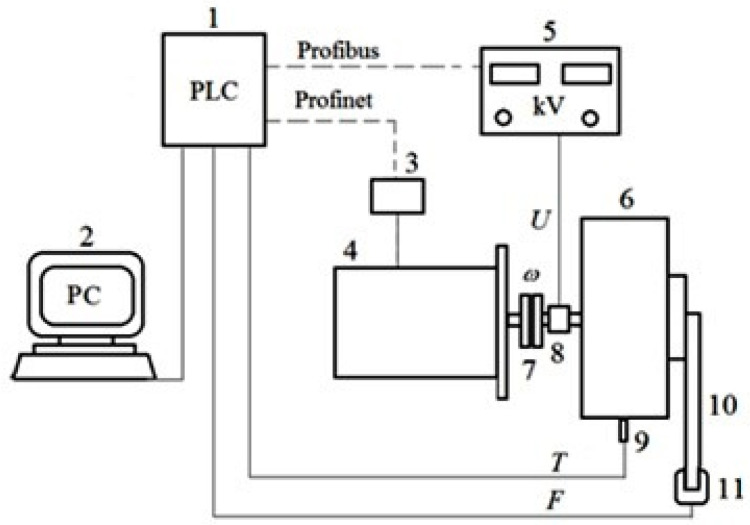
Test bench scheme: 1—PLC controller, 2—PC, 3—inverter, 4—electric motor, 5—high-voltage power supply, 6—viscous brake with ER fluid, 7—connecting coupling, 8—slip rings and brushes, 9—temperature sensor, 10—lever, 11—force sensor.

**Figure 5 sensors-23-06996-f005:**
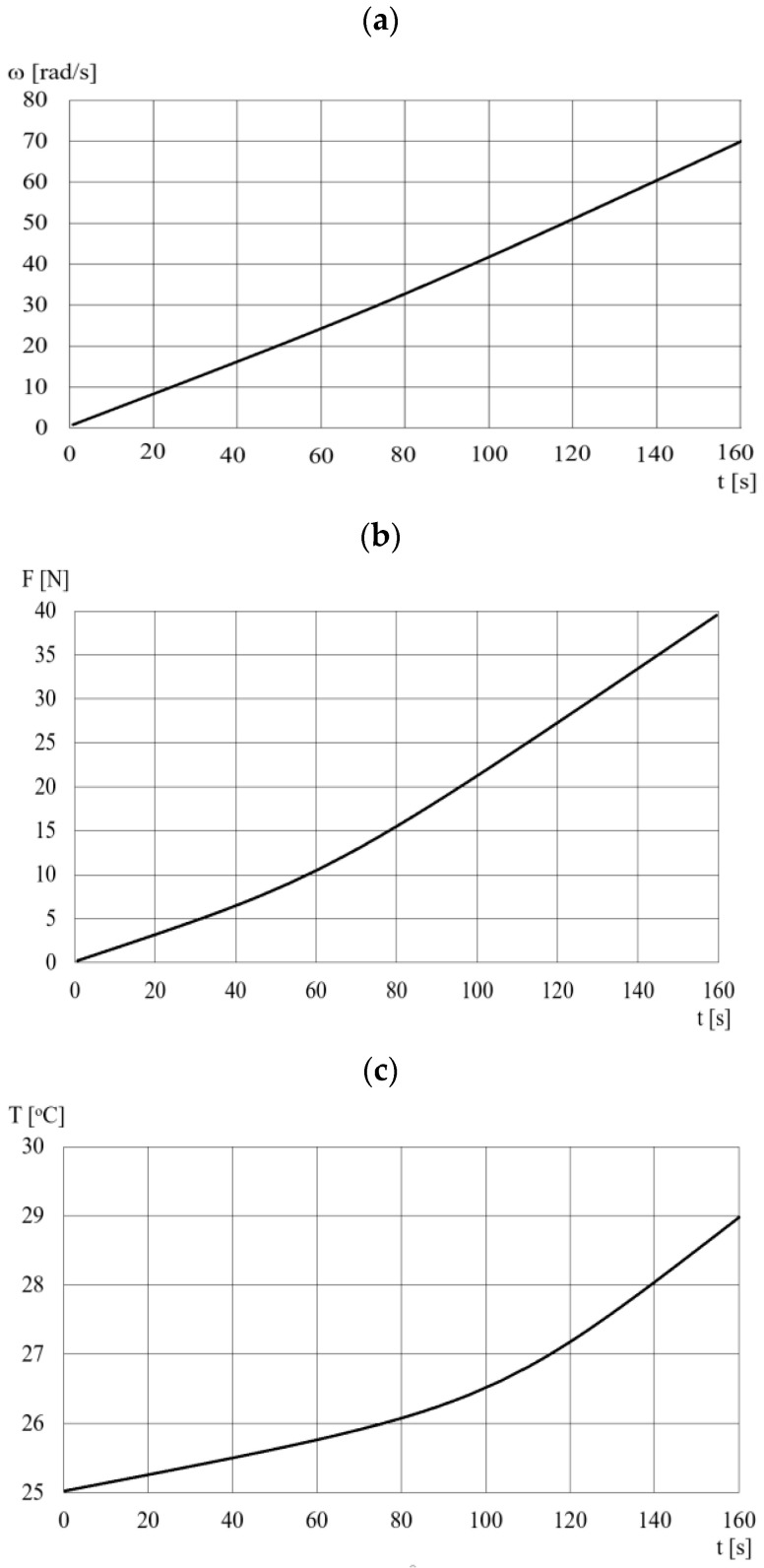
The course of changes over time *t* of: (**a**)—angular velocity *ω*, (**b**)—pressing force *F*, (**c**)—temperature *T*.

**Figure 6 sensors-23-06996-f006:**
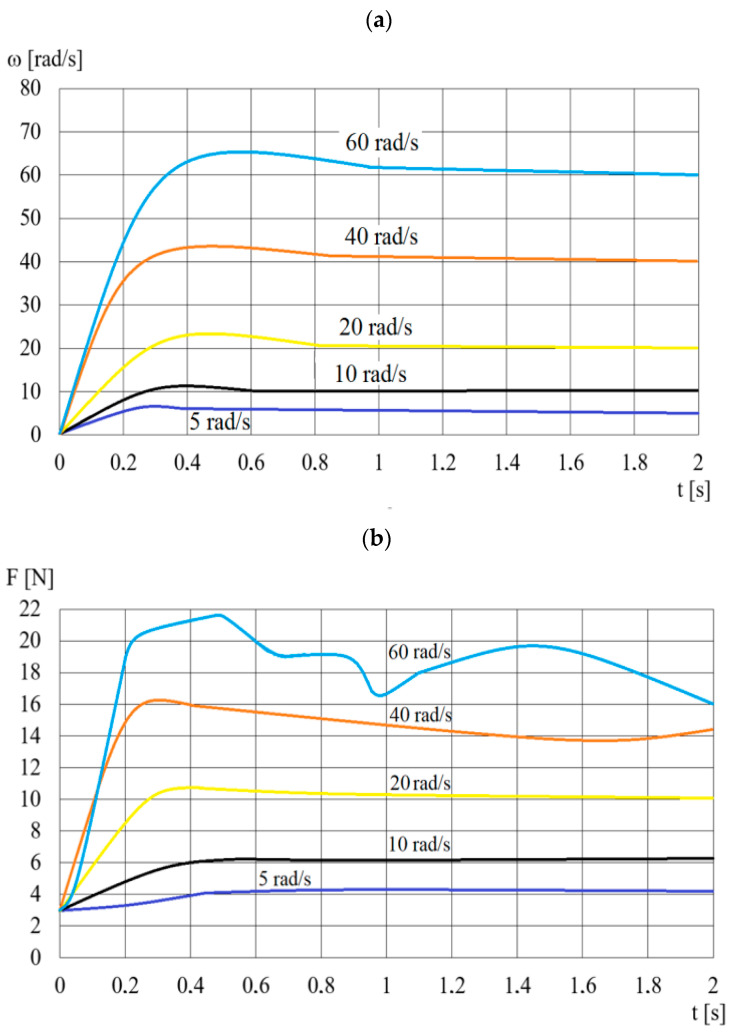
The course of changes over time *t* of: (**a**)—angular velocity *ω*, (**b**)—pressing force *F*.

**Figure 7 sensors-23-06996-f007:**
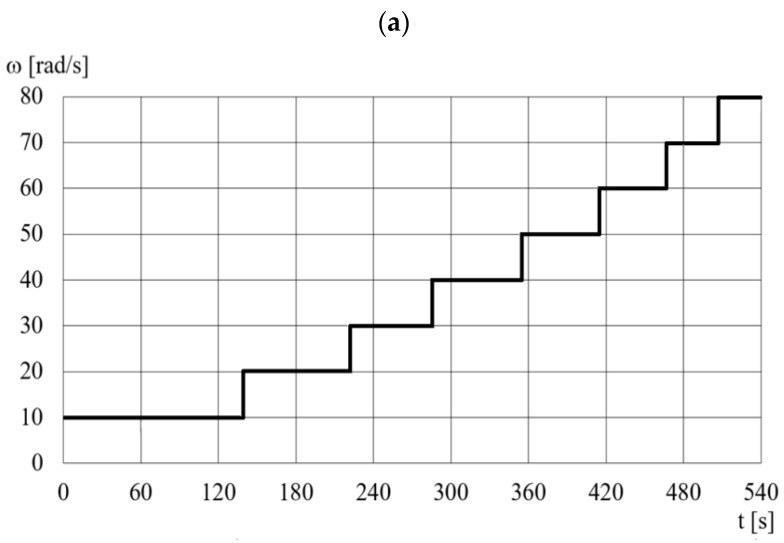
Course of changes in time *t* for temperature *T* = 50 °C of: (**a**)—angular velocity *ω*, (**b**)—high voltage *U*, (**c**)—pressing force Δ*F*.

**Figure 8 sensors-23-06996-f008:**
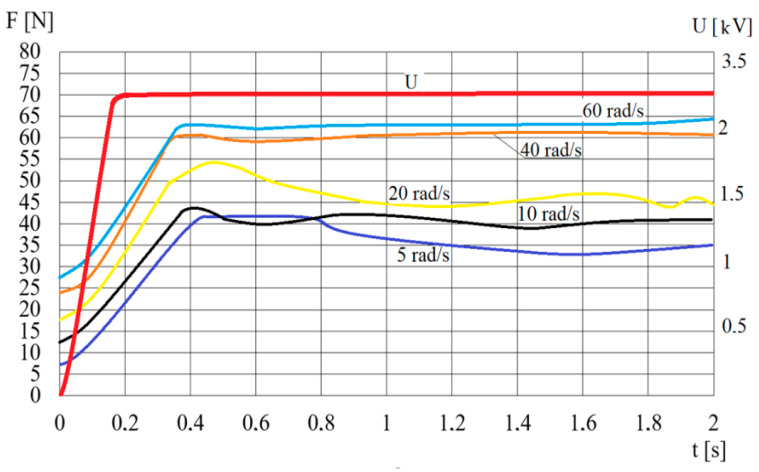
Course of changes in time *t* of the pressing force *F* for a step change in the electric voltage *U*.

**Figure 9 sensors-23-06996-f009:**
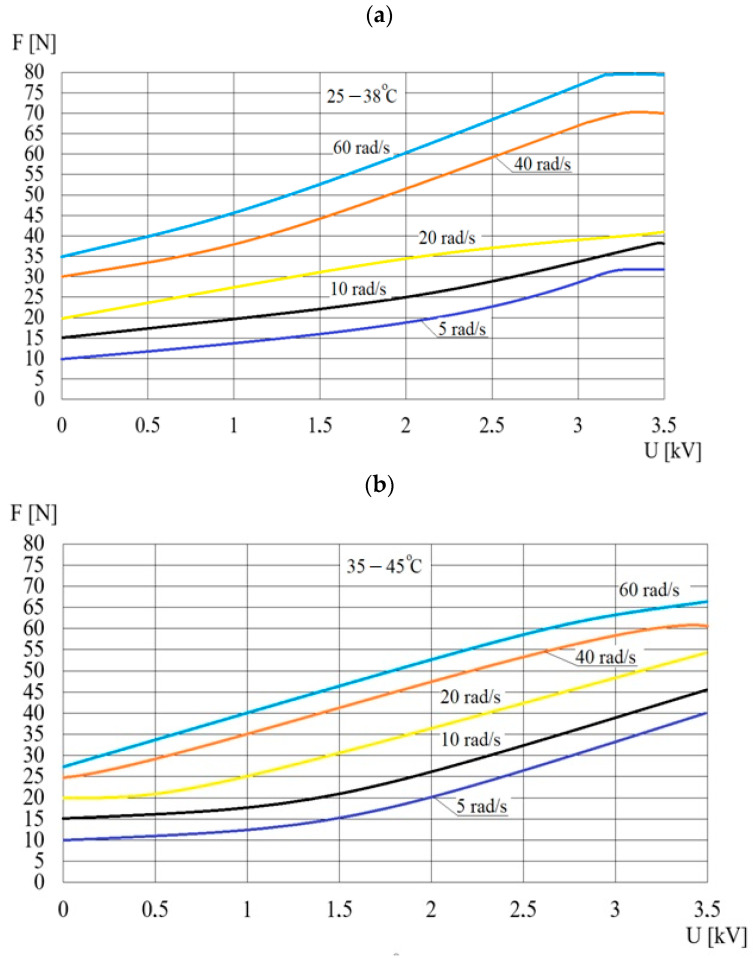
The dependence of the pressing force *F* on the electric voltage *U* for the temperature ranges: (**a**)—25 °C to 38 °C, (**b**)—35 °C to 45 °C.

**Figure 10 sensors-23-06996-f010:**
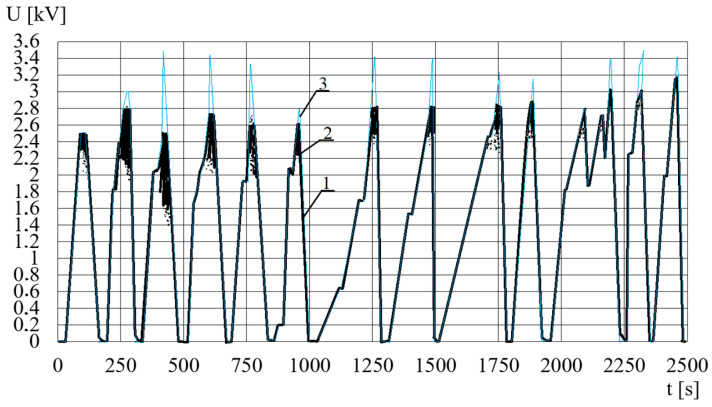
The course of the voltage *U* in time *t*: 1—correct, 2—electric breakdowns, 3—desired electric voltage.

**Figure 11 sensors-23-06996-f011:**
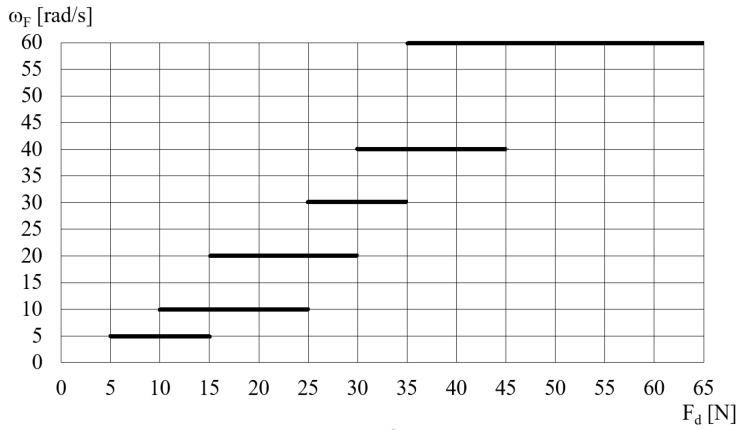
The dependence of the angular velocity *ω_F_* on the desired pressing force *F_d_*.

**Figure 12 sensors-23-06996-f012:**
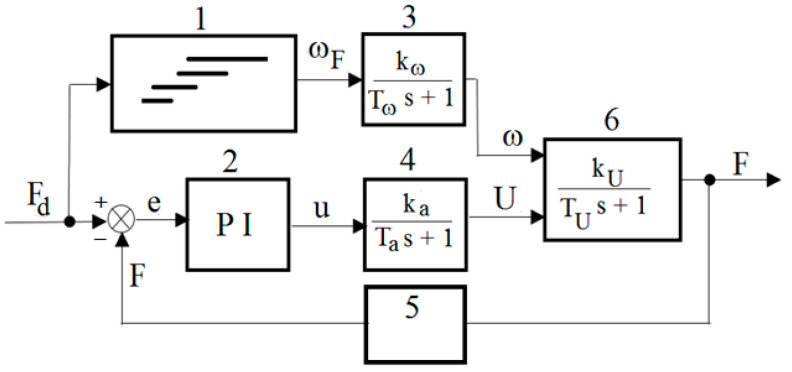
Control system scheme of DECPF: 1—dependence of *ω_F_* on *F_d_*, 2—PI controller, 3—electric motor, 4—high-voltage power supply, 5—force sensor, 6—brake with the ER fluid, *F_d_*—reference torque, *e*—resultant error, *u*—control signal, *U*—control voltage.

**Figure 13 sensors-23-06996-f013:**
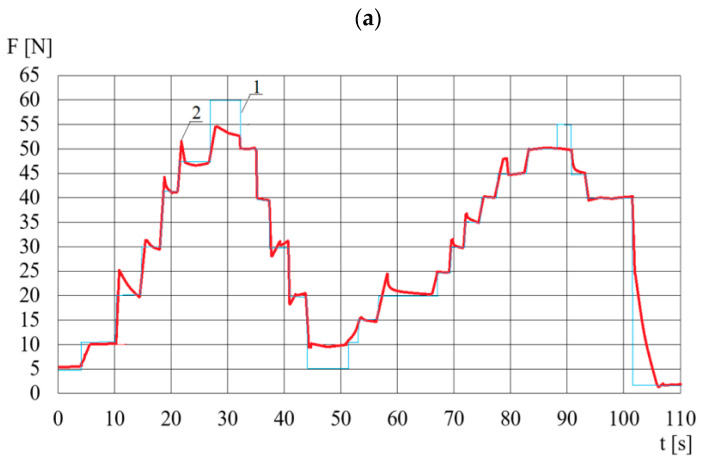
Course of changes in time *t* when controlling: (**a**)—pressing force *F*, 1—desired force, 2—measured force; (**b**)—angular velocity *ω*; (**c**)—high voltage *U*.

**Figure 14 sensors-23-06996-f014:**
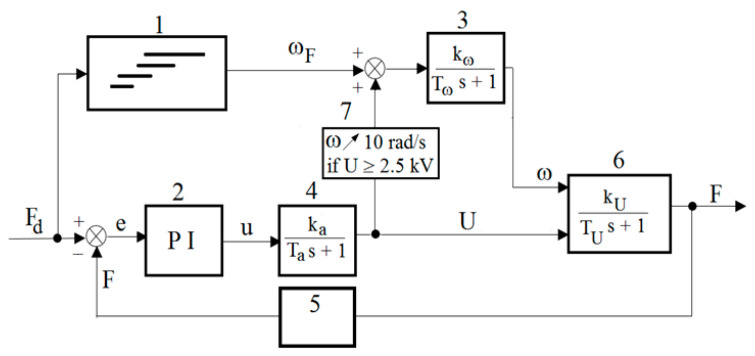
Corrected control system scheme of DECPF: 1—dependence of *ω_F_* on *F_d_*, 2—PI controller, 3—electric motor, 4—high-voltage power supply, 5—force sensor, 6—brake with the ER fluid, 7—correction block, *F_d_*—reference torque, *e*—resultant error, *u*—control signal, *U*—control voltage.

**Table 1 sensors-23-06996-t001:** DECPF design data.

Parameter	Unit	Value
Outer radius	mm	120
Width	mm	110
Number of cylinders in the driving part	-	3
Number of cylinders in the driven part	-	2
Number of working gaps	-	5
Width of the gap between the cylinders	mm	1.0
Lever length	mm	250
ERF#6 fluid volume	cm^3^	395

**Table 2 sensors-23-06996-t002:** Basic data of the ERF#6.

Parameter	Unit
Solid phase	Sulfonated resin
Size of solid particles	10 μm
The base oil	Silicon oil
Dynamic viscosity at 20 °C	*μ* = 65 mPa·s
Density	*ρ* = 1.074 g cm*^−^*^3^
Share of the solid phase by volume	*φ_o_* = 35%
Yield stress at 2.5 kV and 20 °C	*τ*_0_ = 1.8 kPa

**Table 3 sensors-23-06996-t003:** The electrical and electronic components used in the construction of the test bench.

Type	Model	Producer
PLC	6ES7-151-8AB01-0ABO	Siemens
Electric motor driver—Inverter	Unidrive SP 14 × 06	Emerson Industrial Automation
Invertercommunicationcard	SM-Profinet	Emerson Industrial Automation
Inductive motor	3SKg 132-4, 5.5 kW	Tamel
PC computer	COMPAQ DC7900	Compaq
High-voltage power supply	HCP 3500-350	Fug
Force sensor	KMB19-K-100N 0000-D	P.P.H. Wobit E.K.J.
Temperature sensor	Heraeus M222	Conrad Electronic

**Table 4 sensors-23-06996-t004:** Identified *T_ω_* and *k_ω_* for a step change of angular velocities.

*ω* (rad/s)	Maximum *F* (N)	*T_ω_* (s)	*k_ω_* (N/(rad/s))
5	4.56	0.10	0.90
10	6.79	0.20	0.70
20	10.85	0.31	0.53
40	16.98	0.30	0.35
60	21.70	0.30	0.30

**Table 5 sensors-23-06996-t005:** Increase in pressing force Δ*F* for different angular velocities *ω*.

*ω* [rad/s]	*F* [N] for 0 kV	*F* [N] for 3.5 kV	Δ*F* [N]	Δ*F* [%]
10	5.82	38.05	32.23	554
20	10.62	45.93	35.31	332
30	13.37	49.05	35.68	267
40	16.99	45.72	28.73	169
50	20.67	54.91	34.24	166
60	24.99	58.23	33.24	133
70	29.81	58.81	29.00	97
80	33.12	62.21	29.09	88
90	36.34	61.33	24.99	69

**Table 6 sensors-23-06996-t006:** The identified *T_U_* and *k_U_* for a step change in electric voltage from 0 to 2.5 kV.

*ω* (rad/s)	Maximum *F* (N)	*T_U_*	*k_U_* (N/kV)
5	40.23	0.29	14.0
10	43.17	0.30	16.2
20	52.82	0.24	18.0
40	59.11	0.25	24.0
60	62.06	0.22	25.2

**Table 7 sensors-23-06996-t007:** Breakdown voltage *U_b_* values and pressing force *F* values for 90% filling.

*ω* (rad/s)	*U_b_* (kV)	*F* (N)
5	2.4	28.28
10	2.4	49.12
15	2.5	48.25
20	2.5	50.64
25	2.5	50.35
30	2.6	61.11
35	2.7	65.35
40	2.7	60.91
45	2.8	57.36
50	2.8	58.02
55	2.8	52.23
60	3.4	63.87

**Table 8 sensors-23-06996-t008:** PI controller settings.

Method	*k_P_*	*k_I_*
Ziegler–Nichols firstmethod	0.02	0.0337
Ziegler–Nichols secondmethod	0.09	0.1355
Manual setting correction	0.05	0.2000

## Data Availability

Not applicable.

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
