# Peer review of "A Feedback Control Sensing System of an Electrorheological Brake to Exert a Constant Pressing Force on an Object"

_sensors, 2023, doi:10.3390/s23156996_

Round 1
Reviewer 1 Report
The manuscript reports on feedback control sensing system of an electrorheological (ER) brake. The authors provide short introduction in the field of ER fluids, and adequate review of ER fluids clutches and brakes design and control. The effects of angular frequency, temperature of fluid, and electric field strength on the pressing force of cylindrical viscous brake with lever are considered. A feature of the work is the analysis of the mentioned parameters with incomplete filling of the brake (up to 90%). Two control system schemes with feedback are proposed and analyzed. Due to its particular scientific interest to science community, I would recommend this paper for its publication to Sensors.
However, there are some questions and comments for the authors to consider:
1) The definition of ER fluids reflective the properties of the material, namely the change in viscosity under the action of an electric field, but not just the typical composition, should be given in section 2.1. Otherwise, it will be difficult for a reader unfamiliar with the concepts of MR and ER fluids to comprehend the principle of ER brakes operation, which are discussed below.
2) The ER characteristics of the ERF#6 fluid, for example, the values of the yield stress (or viscosity) in an electric field should be given in Table 2 as well.
3) How did the driving and driven parts of the brake isolated from each other (line 234)?
4) The results in Figure 6c are not clear to me. Is the increase in temperature over time observed for a constant or increased angular velocity? I suppose that the graph provided for increased angular velocity according Figure 6a. If my assumption is correct, then how will the temperature change in time at a constant angular velocity?
5) The abrupt change of pressing force at 60 rad/s should be discussed in more details (Figure 7b).
6) I’m slightly confused by results presented in Table 7. The pressing force values for 90 % brake filling are higher than values shown in Tables 5 and 6.
Some other minor comments:
7) The mention of ERF#6 ER fluid in Section 2.2 (lines 134-135) is confusing as the description of the fluid is only given in Table 2, Section 3.
8) Figure 4 is superfluous in my opinion and can be combined with Figure 2 or Figure 3.
9) The list of references includes less than 30% of sources from the last 5 years.
10) There are several typos that should be fixed:
extra space, lines 93, 171, 198 etc;
"gap" instead of "bob" or "cylinder", line 108;
"firs" instead "first", line 156;
different symbols (italic and not) used to represent high voltage, lines 289-290;
"liquid" instead "fluid", line 291;
space is missed, line 376;
"thriugh" instead "through", line 418.
Minor editing of English language required.
Author Response
The authors' reply has been attached.

Reviewer 2 Report
In the introduction, authors should discuss about the disadvantages of DECPF systems and their long term reliability. Instead of literature overview which is usually followed in project reports/thesis, it can be included in the introduction. The literature are discussed like a project report and should be discussed like a journal paper. No need to discuss things in detail , it is better to discuss the findings only.
Information about where the ER fluid was procured is missing.
Instrumentation, sensors and data acquisition systems are also not given.
Line 384-"It is also estimated 383 that the time constant Ta of the electric power supply is 0.18 s for a voltage spike of 2.5 kV,"- how was this estimated?
Line 389-90- " determined for two temperature ranges: from 25 °C to 38 °C, and from 35 °C to 45 °C. " - Why was it limited to those temperatures? What happens beyond that temperature range?
Important detail about how much of an influence the viscosity of the fluid have on the results should be explored since the study involves Electro-Rheological fluid.
Line 463-64-" In order to limit the inertia effect, it is assumed that the change in angular 463 velocity from one selected value to another occurs with an acceleration of 20 rad/s2" what is the basis for the assumption?
What is the robustness of the developed control system?
Not much information about the development of the control system is mentioned.
Apart from these the following observation have also been made about the paper:
1.Quantification of the work is necessary in the abstract
2. The novelty of the work is not clear and should be mentioned in the abstract and in the results.
3. The method in which citations are written is not accordance with a journal paper format, for example- Line 63, 77, 107,113,129, 137,178-" The publication [11] mentions" , this type of representation should not be used. instead it can be written as First author et. al reported...
4.Line 402- instead of 250 C -380C, better to write 250 C to 380C
5.Line 500- what is "The scheme od thus"?
6.line 418- leakage current ig flowing thriugh- spelling mistake.
The above said comments should be addressed.
The language needs lot of improvement and authors should use tools like whitesmoke to refine the write up. The grammatical and spell check also is recommended.
Author Response
The authors' reply has been attached.

Round 2
Reviewer 2 Report
The paper titled " A feedback control sensing system of an electrorheological 2
brake to exert a constant pressing force on an object" is an interesting work and the authors have made the changes as suggested.
But some minor changes are needed which is listed below:
Line 88-n the article [27]- instead give the name of the author of the paper.
Line 120-In the publications [34, 35]-instead give the name of the author of the paper.
Otherwise the changes are acceptable.
The quality of English has been modified and improved.
